# Blood and Salivary Inflammatory Biomarkers Profile in Patients with Chronic Kidney Disease and Periodontal Disease: A Systematic Review

**DOI:** 10.3390/diseases10010012

**Published:** 2022-02-17

**Authors:** Lisandra Taísa Reginaldo Tavares, Mariana Saavedra-Silva, Joaquín Francisco López-Marcos, Nélio Jorge Veiga, Rogerio de Moraes Castilho, Gustavo Vicentis de Oliveira Fernandes

**Affiliations:** 1Departamento de Cirurgía (Área de Estomatología), Facultad de Medicina, Universidad de Salamanca, 37007 Salamanca, Spain; lisandrataisa@hotmail.com (L.T.R.T.); jflmarcos@usal.es (J.F.L.-M.); 2Biomedicine at the Medical Science Department, University of Beira Interior, 6201-506 Covilhã, Portugal; marianasaavedrasilva86@gmail.com; 3Center for Interdisciplinary Research in Health (CIIS), Facultade de Medicina Dentária, Universidade Católica, 3504-505 Viseu, Portugal; nelioveiga@gmail.com; 4Periodontics and Oral Medicine Department, University of Michigan, Ann Arbor, MI 48109, USA; rcastilh@umich.edu

**Keywords:** chronic kidney disease, periodontitis, biomarkers, systematic review, proteins

## Abstract

**Introduction:** Periodontitis is the most prevalent inflammatory disease worldwide. Its inflammatory levels spread systemically, which can be associated with chronic kidney disease. Biomarkers have the potential to diagnose and correlate periodontitis and chronic kidney disease, helping to monitor systemic inflammation. Thereby, this study aimed to analyze the association between chronic kidney disease and periodontitis by conducting a biomarker analysis on blood and saliva. **Material and methods:** An electronic search through PubMed/MEDLINE, EMBASE, and Web of Science databases was conducted to identify clinical studies published in the last ten years, with no language restrictions. Twelve articles met all the inclusion criteria, two randomized controlled trials, one cohort study, and nine observational studies. **Results:** The studies included a total of 117 patients for saliva biomarkers, with a mean age of approximately 57 years old, and 56.68% of the subjects were female. After analyzing all the included studies, it was possible to verify the following biomarkers assessed: CRP, WBC, fibrinogen, IL-4 and -6, cardiac troponin T, NOx, ADMA, albumin, osteocalcin, cystatin C, PGLYRP1, cholesterol, HDL, LDL, triglycerides, and hemoglobin. **Conclusion:** A direct cause–effect association between periodontitis and CKD could not be established. However, it was possible to conclude that there was a correlating effect present, through the analyzed biomarkers.

## 1. Introduction

Chronic kidney disease (CKD) consists of a progressive and irreversible change to the normal kidney function and/or damage to the renal parenchyma at the glomerular, tubular, or endocrine level [1,2,3]. It is characterized by the loss of the filtration capacity of the kidneys and the consequent accumulation of organic residues (urea) that cause immunodeficiency due to the increase of toxic substances in the bloodstream. In addition, there is a loss of hormonal production capacity, control of the electrolyte balance, and blood pressure [1,2,3,4].

Moreover, CKD patients have oral manifestations, with radiographic changes in bone density, more significant accumulation of dental plaque [4], bleeding on the oral mucosa associated with alterations in platelet aggregation, and renal anemia [5]. Furthermore, these oral manifestations are related to xerostomia, in which the patients are typically prone to dry mouth and retrograde parotitis. Beyond one-third of hemodialysis patients have halitosis (“uremic fetor”) and experience a metallic taste due to the high content of urea in saliva [4,6]. 

Furthermore, burning mouth syndrome is frequently seen as an additional symptom in dialysis patients [6]. It presents lesions, localized or generalized erythematous areas, covered by pseudomembranous exudates, that leave an intact or ulcerated mucosa [6]. Moreover, studies [6,7] reported that 4% of patients undergoing hemodialysis suffer from angular cheilitis and lichen planus, which can arise in association with antihypertensive medication. 

In addition, CKD positively correlates with periodontal disease viability due to systemic inflammatory burden and low immunity, with a greater predisposition or worsening of the periodontal condition. Furthermore, periodontal disease may be one of the aggravating factors of progression or mortality in CKD patients (bidirectional cause–effect relationship) [1,4,8,9,10,11]. The presence of periodontal disease simultaneously worsens the prognosis due to the increased systemic inflammatory load and bacterial translocation. Still, the treatment and control can act as an adjunct therapy for both [1].

Moreover, cross-sectional studies conducted in different countries reported that severe periodontitis was significantly more frequent, severe, and prevalent among CKD patients [12,13,14,15,16].

However, in periodontal disease and in several chronic systemic diseases, such as CKD, an early diagnosis is essential. Classically, CKD is evidenced by blood and/or urine laboratory tests, or salivary biomarkers [8,17]. The literature has presented crevicular and salivary fluids analyses as laboratory tests to detect periodontitis. Saliva is the biofluid most studied and analyzed by the scientific community [18,19], and is collected in a non-invasive way in order to reduce the patient’s anxiety levels (no needle, which makes it easier and faster) [11,20]. In terms of storage, it is also easier than for blood as saliva does not clot [11,18,21].

Biomarkers exist in various forms belonging to five different fields: genomics, transcriptomics, proteomics, interactomics, and metabolomics. Changes in the concentration, structure, function, or action of the various components can be associated with the onset, progression, or even regression. In this way, salivary biomarkers serve as a valuable and attractive tool in the detection, risk assessment, diagnosis, prognosis, and monitoring of the disease [22,23].

At the salivary level, patients with chronic renal failure, compared to healthy individuals, have an elevated salivary pH, increased systemic inflammatory activity, and higher concentrations of C-reactive protein (CRP), urea, sodium, and potassium, in conjunction with significantly lower calcium values [10,24,25]. Studies have shown that the concentrations of some metabolites in saliva, such as creatinine, urea, and potassium, differ according to the renal failure degree and treatment with dialysis [26].

CKD patients suffer from the suppression of humoral and cellular immune responses, which leads to subnormal immunoglobulin (Ig) A and IgG concentrations [27]. According to studies, it is possible to find changes in the salivary inflammatory biomarkers in patients with CKD by analyzing interleukin (IL) 1β (IL-1β), IL-6, IL-8, tumor necrosis factor-α (TNF-α), interferon-γ (INF-γ), monocyte chemoattractant protein-1 (MCP-1), and intercellular adhesion molecule-1 (sICAM-1) [11]. In addition, the greater part of these cytokines are present in periodontitis, mainly IL-1, IL-6, and TNF-α, which can lead to tissue destruction and the consequent loss of insertion [1,9,11,22,28]. Moreover, other important salivary biomarkers that are increased in periodontitis are metalloproteinase (MMP)-8, MMP-9, and the inflammatory protein of macrophages-1α (MIP-1α or CCL3) [29].

The medical system depends on a correct diagnosis to adequately construct a treatment plan, and the salivary analysis can be relevant in this aspect. It can be applied in many subareas, which can involve kidney disease or periodontitis [11,18,21,22,30,31,32]. Thus, it will allow the clinician to make an informed decision on the diagnosis and treatment choice without wasting hours waiting for classical laboratory exams. Therefore, this systematic review sought to analyze the correlation between CKD and periodontitis within this context, with regards to assessing the existence of and changes in blood and salivary biomarkers.

## 2. Materials and Methods

This systematic review followed the Preferred Reporting Items for Systematic reviews and Meta-Analysis (PRISMA) guidelines. The focal question was determined according to the Population (P), Intervention (I), Comparison (C), and Outcome (O), PICO strategy. The protocol for this systematic review was registered in the PROSPERO platform (Centre for Reviews and Dissemination/CRD—University of York) and, consequently, accepted with the acceptance number CRD42020168324.

### 2.1. Focal Question

The focal question for the present review was as follows: “In clinical studies of patients with chronic kidney disease and periodontitis (P), will the analysis of saliva biomarkers (I), or of blood biomarkers (C), exhibit important outcomes (O)?

### 2.2. Information Sources and Search Strategy

A systematic search of the literature published in the last eleven years was carried out (1 January 2010–31 January 2021). Three bibliographic databases were used to search as thoroughly as possible (PubMed, Web of Science, and Embase). In each search engine, the search was adapted to its characteristics, based on Boolean operators (AND, OR) to combine searches for mesh terms, Emtree terms, and common terms. In PubMed, Web of Science, and Embase, the following terms were used: (“chronic kidney disease” OR “chronic renal disease”) AND (periodontitis OR “periodontal disease”) AND (“inflammatory biomarkers”) AND (“blood biomarkers” OR “salivary biomarkers”). Articles written in any language were included (Table 1). Searches in the reference lists of the included studies (cross-referencing) were also conducted. All used articles were stored in the bibliographic management platform Mendeley.

### 2.3. Inclusion and Exclusion Criteria

Inclusion and exclusion criteria were designed to permit replicability. Only human studies were included. Furthermore, articles from different languages and with less than ten years of publication were analyzed to get the most recent evidence possible.

Exclusion criteria consisted of previously performed systematic reviews and meta-analyses, studies unrelated to patients with CKD and/or periodontitis, and studies that did not address salivary/blood biomarkers. Moreover, studies that did not involve humans, and only analyzed crevicular fluid or urine, were also excluded.

### 2.4. Study Selection

The selection of studies was carried out based on strict selection criteria, date of publication, and target population. In order to be transparent and replicable, the study selection followed the inclusion and exclusion criteria detailed in Table 2.

Duplicate articles were excluded, and the remaining were elected through the initial reading of the title and abstract. The reviewers (L.T.R.T. and S.M.S.R.P.S.) individually screened the papers, and any disagreements between reviewers were discussed with a third reviewer (G.V.O.F.), who made the final decision.

### 2.5. Data Extraction and Assessment of Bias

Included were trials that enrolled patients with CKD and periodontitis for the analysis of salivary/blood biomarkers. Two reviewers extracted the following data: year of publication, study design, participants (population setting), study period, summary characteristics of study participants (age, gender, periodontal disease, CKD stage, HbA1c, smoking habits), biofluid analysis, biomarkers analyzed, and study results.

Assessment of risk of bias and study quality of the included investigations was performed independently by two reviewers (L.T.R.T. and G.V.O.F.), where the STROBE Statement was applied. The guidelines featured 22 items that were answered with one of four options: (1) yes, (2) no, (3) cannot answer, and (4) not applicable. Only items with option (1) generated the score. Therefore, each article could obtain a score of between 0 (no criteria fulfilled) and 22 (all criteria fulfilled).

The data collected using the STROBE statement was rated as low quality if it scored 0–7 out of 18 points, moderate quality if it scored 8–15, or high quality if it scored between 16 and 22. The ratings obtained were verified by a third reviewer (N.J.V.), and any discrepancy was resolved by discussion with another reviewer (J.F.L.M.).

## 3. Results

### 3.1. Study Selection

A total of 7051 articles were identified through database searching, 467 from PubMed, 4221 from Web of Science, and 2363 from Embase. A further 16 publications were considered from the manual search through the references of the included articles. From 7067 articles, 108 were excluded due to duplication. Therefore, 6959 articles were screened by title and abstract based on the inclusion and exclusion criteria (Table 2). After the title and abstract screening, 43 articles were included for further analysis. Subsequently, there was a full-text assessment, in which 31 articles were excluded based on the exclusion criteria detailed in Table 3. Twelve articles [33,34,35,36,37,38,39,40,41,42,43,44] met the inclusion criteria and were included in the current review (Figure 1).

### 3.2. Study Characteristics

Within the 12 studies selected for analysis, only two were randomized controlled trials (RCT), one was a cohort study (CS), and nine were observational studies (OB) (seven out of nine were cross-sectional studies, CSS), all published between January 2010 and January 2021. The studies included a total of 102,000 patients for blood biomarkers and 117 patients for saliva biomarkers, with a mean age of approximately 57 years old (ranging between 23 and 86 years), and 56.68% of the subjects were female.

All the investigations studied the periodontitis and CKD condition of the patients. Regarding the CKD condition, from 12 studies, all patients had a compromised kidney function, varying from CKD stage 1 to end-stage renal disease (ESRD) or required renal replacement therapy (RRT). Only one study used a completely healthy control group [33]. Six studies compared healthy oral patients and patients with different levels of periodontitis. Most of the studies also considered other health markers, such as diabetes and smoking habits (Table 4).

Of the 12 studies, 11 conducted a blood collection, and only one [43] used saliva to obtain the required data. Different types of biomarkers were analyzed, such as metabolism, nutritional, inflammatory, and bacterial biomarkers.

### 3.3. Chronic Kidney Disease (CKD) and Periodontal Disease

Regarding CKD, in eight articles, the subjects required RRT, and most of them were undergoing hemodialysis (HD) [33,34,35,36,37,38,43,45]. Perozini et al. [36] was the only study that managed to examine the patients at all stages of the disease (from CKD stage 1 to stage 5). Chen et al. [44] defined patients with CKD as having an estimated glomerular filtration rate (eGFR) of 68.76 (±17.6 mL/min/1.73 m^2^), which means that patients’ conditions were between CKD stage 2 and stage 3.

Concerning periodontal disease, all studies included patients with periodontitis. Seven out of 12 studies made a comparison between patients with periodontitis and patients without periodontitis [33,34,35,37,38,39,46]. None of the studies classified periodontal disease by the new classification of the World Workshop on the Classification of Periodontal and Peri-implant Diseases and Conditions [47]. Furthermore, one study (Hou et al., 2017) did not distinguish gingivitis from periodontitis, due to their similar pathologies.

### 3.4. Biomarkers

After analyzing all the included studies (Appendix A), it was possible to verify the following biomarkers that were assessed: (i) C-reactive protein (CRP), (ii) white blood cells (WBC), (iii) fibrinogen, (iv) IL-4, (v) IL-6, (vi) cardiac troponin T (TnT), (vii) nitric oxide (NOx), (viii) asymmetric dimethylarginine (ADMA), (ix) albumin, (x) osteocalcin, (xi) cystatin C, (xii) peptidoglycan recognition protein 1 (PGLYRP1), (xiii) cholesterol, (xiv) high-density lipoprotein (HDL), (xv) low-density lipoprotein (LDL), (xvi) triglycerides, and (xvii) hemoglobin. In order to describe these biomarkers and verify the relationships among them, further analysis, presented in Appendix A and Appendix A, was performed.

CRP, an inflammatory biomarker, was studied in seven articles [34,35,36,37,38,40,41], making up 58.33% of the investigated studies. In Cotic et al. [34], 63.1% of the patients with CKD had a CRP higher than 3 mg/L. Therefore, this study did not prove any significant relationship between elevated CRP levels and periodontal disease and observed that early tooth loss resulting from an oral infection was the possible cause of levels higher than 10 mg/L in edentulous patients. Likewise, in Perozini et al. [36], CRP levels were higher than the reference value, although no significance was found. On the other hand, the other articles showed significantly higher levels of CRP in patients with periodontitis. According to Veisa et al. [38], the CKD patient’s periodontitis led to high levels of CRP, decreasing the patients’ quality of life. Grubbs et al. [40] demonstrated that intensive and local periodontal treatment reduces the CRP levels.

One study considered WBC count [37] and reported a significantly elevated quantity of WBCs in CKD patients with periodontitis. Although fibrinogen is also a biomarker of inflammatory activity, only Perozini et al. [36] incorporated it into their analysis. They found that fibrinogen appeared to be higher in the predialysis patients with periodontitis, although, despite this, they were still within the reference values. The article by Ksiazek et al. [33] showed that changes in 3VNTR polymorphism in the IL-4 gene might be a risk factor in the progression of CKD and periodontitis. Regarding IL-6, another cytokine involved in the inflammatory response and directly linked to periodontitis, Grubbs et al. [40] showed that IL-6 concentration decreased after intensive periodontal treatment.

Cotic et al. [34] could not make a direct association between the influence of oral health on the cardiovascular biomarkers that are usually elevated in CKD patients, such as cardiac troponin T (TnT) and nitric oxide (NOx). Conversely, Grubbs et al. [40] found that asymmetric dimethylarginine (ADMA), an endothelial dysfunction marker, decreased after periodontal treatment in CKD patients. Hemoglobin was used as a biomarker by Hou et al. [37] and Chen et al. [44]; however, the values were normal in CKD patients with and without periodontitis in both studies.

Five articles studied albumin as a biomarker [35,38,39,42,44]. Two of them pointed to a possible association between periodontitis in CKD patients and hypoalbuminemia [35,38]. Garneata et al. [35] reported that 93% of malnourished patients (albumin < 3.5 g/dL) had periodontitis. Rodrigues et al. [39] also studied phosphorous concentration and concluded that CKD patients without periodontitis had a higher concentration than patients with periodontitis. Nevertheless, Hou et al. [37] did not find any significance in the variation of phosphorous concentration. Regarding bone metabolism, Yoshihara et al. [41] studied osteocalcin as a biomarker. They found that osteocalcin in patients with CKD had a significant positive relationship with periodontitis, in which the worst periodontal condition meant higher levels of osteocalcin.

Cystatin C is a biomarker of kidney function; only one article [41] used it to define kidney function (cystatin C > 0.91 mg/L indicates poor kidney function). The authors found a significant positive association between serum cystatin C and periodontal disease, concluding that patients with decreased kidney function had a higher probability of periodontal disease.

Arenious et al. [43] conducted a sequential study based on four previously published studies. This was the only study in the present analysis that collected saliva as a source for analysis. In this study, the authors found that complications related to peritoneal dialysis had no relationship with periodontal disease. Still, complications related to HD were associated with *Staphylococcus aureus* and, consequently, with high levels of peptidoglycan recognition protein 1 (PGLYRP1).

Total cholesterol, and high-density lipoprotein (HDL) and low-density lipoprotein (LDL) levels were studied in two of the twelve articles [37,39]. Only Hou et al. [37] found a statistical significance for total cholesterol between CKD patients with or without periodontitis, documenting higher levels in the periodontitis group. In Chen et al. [44], HDL and total cholesterol were statistically lower in the non-periodontitis group. Triglycerides were also used as a metabolic biomarker [36,37,44]. Significantly high levels of triglycerides in predialysis and HD patients with periodontitis were reported by Perozini et al. [36]. On the other hand, Hou et al. [37] and Chen et al. [44] did not find any significance in the variation of triglyceride levels.

Overall, our study results indicate that periodontal disease is highly prevalent in CKD patients, and that there is a relationship between them [33,34,35,36,37,38,39,40,41,42,43,44,46]. Yoshihara et al. [41] concluded that CKD and periodontitis could have a reciprocal effect. In three studies, the authors discovered that an interdisciplinary treatment must be done in order to improve oral and systemic health [36,39,40]. Chen et al. [44] and Cotic et al. [34] added that periodontal disease increased the cardiovascular mortality risk in CKD patients, while Veisa et al. [38] concluded that the presence of periodontal disease in CKD patients reduced their quality of life.

### 3.5. Other Health Risk Factors (Smoking and Diabetes)

Although several studies have taken into account smoking habits and diabetes, only two studies found significant results [35,37]. These studies also reported that patients with diabetes and that smoked had increased probabilities of having periodontal disease.

## 4. Discussion

This systematic review analyzed the correlation between CKD and periodontitis with regards to blood and salivary biomarkers, which permitted the inclusion of 12 articles. To the best of our knowledge, this is the first systematic review that evaluates the current evidence for the association between chronic kidney disease and periodontitis by analyzing either saliva or blood as biomarkers. The included articles were led by different principles; however, they all examined the relationship between periodontal disease and CKD. Unfortunately, some studies did not distinguish periodontitis from gingivitis, which may lead to bias. Furthermore, it is important to highlight the scientific evidence that showed that diabetes and smoking habits increased the possibility of having periodontal disease [48,49,50]. Moreover, it is important to emphasize that treating periodontitis in a CKD patient will reduce the systemic inflammation level beyond improving the oral condition.

CRP is a well-known inflammatory protein, and it can be found in the blood and the saliva. This protein influences systemic diseases, such as CKD, and oral infections, such as periodontitis [51,52,53]. Although two studies did not find statistically significant results for this biomarker, other studies reported that patients with CKD and periodontitis had higher levels of CRP, causing a high level of chronic inflammation, with undesirable consequences for the patients’ health [26,27].

Romandini et al. [54] concluded that periodontitis influences WBC count by an increase in systemic inflammation, and Arai et al. [55] concluded that WBC is independently associated with CKD progression. These two articles that studied WBC showed high levels in patients with periodontitis, suggesting a systemic inflammation caused by periodontitis [37,46,56,57].

Similar to CRP, fibrinogen is a biomarker of inflammatory activity [36]. The study that analyzed fibrinogen as a biomarker discovered that values were higher in patients with periodontal disease; otherwise, the values were within the reference values [36]. Moreover, IL-4 and IL-6 are important cytokines in systemic inflammation, and studies have demonstrated that high levels of these cytokines are present in patients with periodontitis [11,58]. Further studies regarding proinflammatory cytokines must be conducted.

Asymmetric dimethylarginine (ADMA) is one of the endothelial disfunction markers responsible for atherosclerosis in CKD patients [59]. One study showed a decrease in ADMA after periodontal treatment, demonstrating that periodontal therapy can be essential to maintain systemic health [60].

Hypoalbuminemia is considered a strong predictor of an adverse prognosis in CKD patients [61]. The article that analyzed albumin concentration found that CKD patients with periodontitis had lower albumin levels, indicating that their health was at risk [62]. Another study regarding albumin levels supports these results and found that there might be an inverse relationship between periodontal disease and serum albumin concentration [63].

Phosphorous concentration was taken into consideration in one of the included studies. Individuals with higher a phosphorous concentration tended to more easily develop dental calculus [64]. Therefore, it was expected that CKD patients with periodontitis had higher phosphorus concentrations; however, studies reported that periodontitis had a lower concentration of phosphorus. These results would mean a positive correlation, in which having periodontitis would lead to the CKD patient having a low phosphorus concentration and, consequently, to better health of the patient. It is known that phosphorus concentration reduction is particularly important to prevent bone impairment and CVD in patients with CKD [65,66]. Rodrigues et al. [39] had some limitations in their study and proposed the development of new studies will enable us to draw accurate conclusions.

One of the studies found that osteocalcin level had a significant positive relationship with periodontitis in CKD patients, with the worst periodontal condition correlated to a higher level of osteocalcin [41]. Conversely, a recently published study concluded a nonsignificant correlation between osteocalcin level and periodontitis severity [67]. Osteocalcin is a bone-forming biomarker, and periodontitis is a chronic disease characterized by the loss of periodontal attachment; thereby, there is a contrast between the effects associated with periodontitis and the presence of higher osteocalcin levels. However, higher osteocalcin levels in CKD meant a higher osteoclast activity, which may lead to bone loss [68].

A significant positive association between the serum cystatin C and periodontal disease was found in one study. These results supported a previous study showing that the level of cystatin C could be used as a marker in chronic periodontitis [41,69]. In addition, increased levels of cystatin C are associated with decreased kidney function, meaning that in the presence of both diseases (periodontitis and CKD), cystatin C was higher, and this can lead to the worst periodontal condition and kidney function [70].

HD complications were related to *Staphylococcus aureus* and, consequently, with high levels of peptidoglycan recognition protein 1 (PGLYRP1) [6], showing a possible relationship between HD complications and oral infection. PGLYRP1 functions as a ligand that activates TREM-1 during illness, so the high values of this biomarker indicate that the patient is probably suffering from systematic inflammation due to periodontitis [71,72,73,74]. Higher values of this biomarker may lead to, or worsen, the prognosis of several diseases, such as rheumatoid arthritis, systemic lupus erythematosus, inflammatory bowel disease, type 1 diabetes, psoriasis cystic fibrosis, sepsis, and even atherosclerotic complications, the latter being one of the leading causes of mortality in patients with CKD [71,72,75].

The results related to total cholesterol, and high-density lipoprotein (HDL) and low-density lipoprotein (LDL) levels, proved subjective and insufficient, as they were contradictory. It is crucial that these biomarkers are studied further, as high concentrations may increase the risk of atherosclerosis, the leading cause of death in CKD patients [10,76,77].

Combining the present study’s conclusions, periodontal disease is highly prevalent in CKD patients, and there is a possible cause–consequence relationship between them [33,34,35,36,39,40,41,43,46]. Interdisciplinary treatment is essential and must be delivered in order to improve oral and systemic health [36,39,40,46]. Cardiovascular disease is one of the main causes of mortality in CKD patients, and periodontitis can influence the prognosis [34,46]. Furthermore, it was found that the presence of periodontal disease in CKD patients lowered their QoL [38].

Finally, the limitations of this study involved the low number of included articles; thus, analysis was confined to the limited data obtained, indicating considerable heterogeneity. Moreover, the lack of standardization regarding the types of proteins analyzed made it difficult to conduct comparisons among studies.

## 5. Conclusions

Within the limitations of this systematic study, it was still possible to realize its importance, due to the systemic inflammation caused by the increase in the proinflammatory response in the presence of periodontitis. The included articles were not enough to establish a direct cause–effect association between periodontitis and CKD, or vice-versa, although promising results were found regarding the biomarkers linking both diseases. Thereby, it was concluded that periodontal treatment therapy can help CKD patient’s prognoses and improve their quality of life, and it must be taken into consideration.

More clinical studies analyzing biomarkers should be performed in order to achieve more predictable and accurate prognoses during the treatment of patients with CKD by controlling the persistent systematic inflammation caused by periodontal disease, first using basic periodontal treatment and targeted therapies.

## Figures and Tables

**Figure 1 diseases-10-00012-f001:**
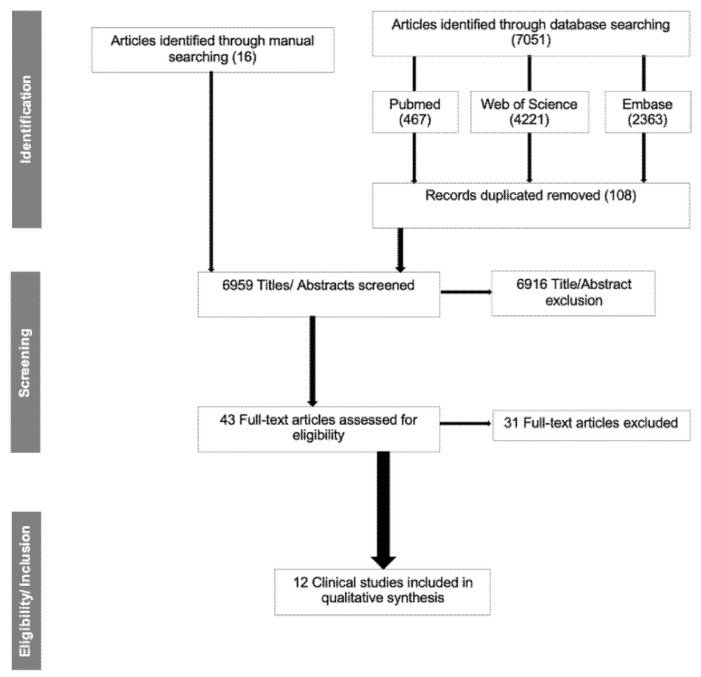
Flow diagram for the search strategy and selection process.

**Table 1 diseases-10-00012-t001:** Search strategy carried out and filters applied.

	MEDLINE (PubMed)	Embase	Web of Science
**#1**	**P—Patients with chronic kidney disease and periodontitis**		
	“Chronic Kidney Disease” [MeSH Terms] AND Periodontitis [MeSH Terms] OR “periodontal disease” [All Terms]	“Chronic Kidney Disease” AND Periodontitis OR “periodontal disease”	“Chronic Kidney Disease” AND Periodontitis OR “periodontal disease”
**#2**	**I—Analysis of saliva biomarkers**		
	…AND “Inflammatory biomarkers” [MeSH Terms] OR “Salivary biomarkers” [All Terms]	…AND “Inflammatory biomarkers” OR “Salivary biomarkers”	…AND “Inflammatory biomarkers” OR “Salivary biomarkers”
**#3**	**C—Analysis of blood biomarkers**		
**#4**	**O—Exhibit similar outcomes**		
**Search Combination**	(**#1** AND **#2**) No combination was done with **#3** and **#4**. Using the **#3** we would be repeating the terms of **#1** and **#2**, and the combination using keywords related to outcome **#4** would limit even more the search and eliminate some relevant studies.		
Filters	Humans, 11 years		

**Table 2 diseases-10-00012-t002:** Inclusion and exclusion criteria used for selection of the articles.

Inclusion and Exclusion Criteria	
Inclusion	Exclusion
Clinical Study	Systematic review/Meta-analysis
Clinical Trial	Review
Clinical Trial Protocol	No biomarkers analyzed
Clinical Trial, Phase I	Blood biomarkers
Clinical Trial, Phase II	Crevicular fluid biomarkers
Clinical Trial, Phase III	Patients without CKD
Clinical Trial, Phase IV	Patient without periodontitis/ peri-implantitis
Research study	Urinary biomarkers
Randomized Controlled Trial	Animal study
Controlled Clinical Trial	
Published in the last 10 years	
Humans	
Any language	

**Table 3 diseases-10-00012-t003:** Excluded studies and reason for exclusions.

Author/Year	Reason for Exclusion
Maciejczyk et al., 2018	Patients with periodontitis were excluded
Joseph et al., 2011	No CKD patients
Kovalciková et al., 2019	
Marinoski et al., 2019	
Schmalz et al., 2017	
Schmalz et al., 2016	
Grubbs et al., 2015	
Caglayan et al., 2015	
Oyetola et al., 2015	No biomarkers were analyzed
Zhao et al., 2014	
Doscas et al., 2018	
Sharma et al., 2014	
Garneata et al., 2014	
Ibrahim et al., 2020	
Iwasaki et al., 2019	
Lertpimonchai et al., 2019	No biomarker association
Machowska et al., 2016	
Opatrná et al., 2015	Review
Hickey et al., 2020	
Demoersman et al., 2018	No clear data for CKD patients
Rodrigues et al., 2016	
Pallos et al., 2015	No direct relation with periodontitis
Luo et al., 2013	
Nylund et al., 2015	Data was updated in 2017, Nylund et al., 2017
Nylund et al., 2017	Data was updated in 2020, Arenius et al., 2020
Trivedi et al., 2018	Trial still recruiting patients
Grubbs et al., 2017	Planning of trial phase, no results
Jamieson et al., 2015	
Sharma et al., 2016	Urinary samples were used instead of blood or salivary samples
Yoshihara et al., 2016	Same results as Yoshihara et al., 2016, with the addition of genotype analysis

**Table 4 diseases-10-00012-t004:** Description of the factors investigated in the included studies.

Study			Patient				Health			
Author/Year	Study Design	Mean Period (Months)	N	Age (Years)		Gender (Female %)	Periodontal Condition	CKD	Diabetes (%)	Smoking (%)
				Mean	Range					
** Blood collection **										
**Cotic et al., 2017 [34]**	RCT	NS	111	NS	26–90	38.74	64.6% had a CPITN > 3	HD	34.2	15.3
**Hou et al., 2017 [37]**	OB	4	135	50.8	50.8 ± 13.3	41.91	51.50% gingivitis/periodontitis	HD	36	55.7 (of 51.50)
**Veisa et al., 2017 [38]**	CSS	31	101	52.5	52.5 ± 14.3	56.44	75.2% periodontitis	HD	5.94	13.86
**Garneata et al., 2014 [35]**	CSS	NS	238	57.0	NS	39.91	75.63% periodontitis	HD	10.92	60.92
**Ksiazek et al., 2019 [33]**	CSS	96	442	63	63 ± 16.3	48.9	40.72% periodontitis	ESRD	NS	NS
**Perozini et al., 2017 [36]**	CSS	16	102	54.74	54.74 ± 13.01	Early stage 75.0; Predialysis 50.0; Hemodialysis 41.7	100% Periodontitis	44 CKD 1-2; 30 CKD 3-4; 28 CKD; 5 (HDG)	Early stage: 40.0; Predialysis: 30.0; Hemodialysis: 16.7	NS
**Rodrigues et al., 2014 [39]**	CSS	6	96	39.8	39.8 ± 13.2	53.13	59.4% periodontitis	HD	NS	NS
**Chen et al., 2015 [44]**	CS	120	100,263	NS	>65	49.1	13.7% periodontitis	eGFR (mL/min/1.73 m^2^) 68.76 ± 17.6	13.9	8.7
**Grubss et al., 2019 [40]**	RCT	12	51	59	34–73	32.3	100% Periodontitis	71% CKD 3 and 29% CKD 4	47.0	10.0
**Yoshihara et al., 2016 [41]**	CSS	NS	332	NS	55–74	100	Periodontitis (NS)	CKD (NS)	NS	No
**Ausavarungnituna et al., 2016 [42]**	CSS	12	129	NS	30–86	24	Periodontitis (NS)	27% severe CKD, 37% moderate CKD and 36% mild CKD	NS	NS
** Saliva collection **										
**Arenius et al., 2020 [43]**	OB	180	117	NS	23–83	29.1	PIBI = 6 (median) TDI = 3 (median)	100% CKD stage 4 and 5 (dialysis)	NS	NS

CKD = chronic kidney disease; CS = cohort study; CSS = cross-sectional study; eGFR = estimated glomerular filtration rate; ESRD = end-stage renal disease; HD = hemodialysis; PIBI = periodontal inflammatory burden index; NS = not specified; OB = observational study; RCT = randomized controlled trial; TDI = tissue doppler imaging.

## Data Availability

All original data were obtained from the articles cited in the respective text. No other value was used or added.

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
