# Peer review of "Blood and Salivary Inflammatory Biomarkers Profile in Patients with Chronic Kidney Disease and Periodontal Disease: A Systematic Review"

_diseases, 2022, doi:10.3390/diseases10010012_

Round 1
Reviewer 1 Report
Periodontitis is a chronic inflammatory disease that is characterised by a disturbed balance between the immune and inflammatory response of the host to bacterial infection. The risk factors of periodontal disease include especially bacterial infections and smoking. Chronic bacterial infection induces an inflammatory response, leading to the increased synthesis of proinflammatory mediators, such as cytokines and chemokines, that can cause changes in various organs.
The topic of this review is interesting. Overall, this review is well done.
The authors should describe in Introduction the relationship between periodontal disease and chronic kidney disease in more detail.
Also worth mentioning in the Discussion section are the possibilities of preventing and treating of periodontal disease in patients with chronic kidney disease.
Author Response
Thank you for all comments.
1. The topic of this review is interesting. Overall, this review is well done.
R.: Thank you very much.
2. The authors should describe in Introduction the relationship between periodontal disease and chronic kidney disease in more detail.
R.: Thank you. We inserted a paragraph and 5 references in the intro session.
3. Also worth mentioning in the Discussion section are the possibilities of preventing and treating of periodontal disease in patients with chronic kidney disease.
R.: It was included as requested.
Reviewer 2 Report
REVIEWERS COMMENTS
Title: Blood and salivary inflammatory biomarkers profile in patients with chronic kidney disease and periodontal disease: A systematic review
Abstract:
- Grammatical errors are highlighted and need to be revised.
- Abbreviations are to be expanded.
Introduction:
- Grammatical errors are highlighted and need to be corrected and reframed.
Materials and methods:
- Grammatical errors are highlighted and need to be revised.
- Abbreviations are to be expanded.
Results:
- Grammatical errors are highlighted and need to be revised.
- Table 5- spelling error in table column needs to be corrected and headings need to be added for last 3 columns.
Discussion:
- Abbreviations are to be expanded.
- Limitations and future trends of the study are to be mentioned.
Author Response
Thank you. We appreciate all comments.
1. Abstract: Grammatical errors are highlighted and need to be revised; Abbreviations are to be expanded.
R.: Thank you. The article was completely reviewed (English language).
2. Introduction: Grammatical errors are highlighted and need to be corrected and reframed.
R.: The introduction was completely reviewed (English language).
3. Materials and methods: Grammatical errors are highlighted and need to be revised; Abbreviations are to be expanded.
R.: Thank you. It was corrected as recommended.
4. Results: Grammatical errors are highlighted and need to be revised; Table 5- spelling error in table column needs to be corrected and headings need to be added for last 3 columns.
R.: Thank you. It was corrected as recommended.
5. Discussion: Abbreviations are to be expanded; Limitations and future trends of the study are to be mentioned.
R.: Thank you. The discussion was reviewed.
Round 2
Reviewer 1 Report
The revised version of the manuscript is suitable for publication.
Reviewer 2 Report
The authors have undergone the revision satisfactorily hence the manuscript can go for publication.